# *Bacillus subtilis*-Derived Surfactin Alleviates Offspring Intestinal Inflammatory Injuries Through Breast Milk

**DOI:** 10.3390/nu17061009

**Published:** 2025-03-13

**Authors:** Qi Zhang, Shuang Xie, Qiu Zhong, Xinyue Zhang, Liufang Luo, Qian Yang

**Affiliations:** MOE Joint International Research Laboratory of Animal Health and Food Safety, College of Veterinary medicine, Nanjing Agricultural University, Weigang 1, Nanjing 210095, China; 17701050457@163.com (Q.Z.); shuangxie1012@163.com (S.X.); 18845097817@163.com (Q.Z.); senioplai.cn@gmail.com (X.Z.); minbiluhai1515@gmail.com (L.L.)

**Keywords:** *Bacillus subtilis* extract, surfactin, breast milk, neonatal intestinal development, immune modulation, intestinal inflammatory injuries

## Abstract

Background: Enteric and diarrheal diseases pose a significant threat to infant health, highlighting the importance of immune defenses in early life, especially maternal protection, in establishing a robust gastrointestinal environment. Surfactin, a bioactive peptide from *Bacillus subtilis*, has immunomodulatory properties, yet its influence on offspring via maternal gut interference is not fully understood. This study examines the effects of maternal surfactin consumption on breast milk’s immunological properties and its consequent effects on neonatal intestinal health. Methods: Twenty-eight gravid mice were randomly categorized into two cohorts and were given surfactin or not in drinking water from one week after conception to 21 days postpartum. Cross-fostering experiments were conducted within 12 h after birth. Pups from the surfactin-supplemented dams were fostered and nursed by the control dams, while the pups from the control dams were nursed by the surfactin-supplemented dams. Results: The findings show that the pups from the surfactin-supplemented dams had increased body weight, improved intestinal morphology with longer villus and deeper crypts, the upregulation of genes related to mucins and antimicrobial peptides, and an increase in IgA^+^ and CD3^+^ T cells within the intestinal mucosa. Further, the cross-fostering experiments suggested that the pups nursed by the surfactin-supplemented dams gained more weight, had less intestinal damage, less inflammation, and lower oxidative stress levels induced by *Salmonella typhimurium*, indicating the immunological benefits of surfactin conveyed through breast milk. Additionally, the expression of pro-inflammatory factors, including nitric oxide, TNF-α, IL-1β, IL-6, MCP-1, and ROS, induced by LPS in the macrophages was significantly inhibited with milk from the surfactin-supplemented dam (MSD) treatment. Interestingly, the MSD treatment induced a shift in macrophage polarization from pro-inflammatory (M1-like) to anti-inflammatory (M2-like), evidenced by the decreased expression of IL-12p40 and iNOS and the increased expression of CD206, TGF-β, and Arg-1. In terms of mechanism, surfactin improved the contents of the anti-inflammatory factors IL-4, IL-10, and TGF-β in the breast milk. Conclusions: This research contributes to understanding how maternal interference can modulate breast milk composition, influence infant gastrointestinal development and immunity, and provide nutritional strategy insights.

## 1. Introduction

Enteric and diarrheal diseases are among the most significant threats to childhood health globally, causing nearly 1 million deaths annually among children under the age of five [1,2]. The intestinal mucosa of newborns, critical for nutrient absorption and pathogen exclusion, is particularly susceptible to inflammatory responses and tissue damage from enteric infections [3,4]. Intestinal damage often occurs during the weaning period, a critical time for gastrointestinal development and immune maturation. In mice, this period typically occurs around postnatal day 21, while in humans, it corresponds to approximately from 4 to 6 months of age [5]. During this transition, the gut undergoes significant physiological changes, making it more susceptible to inflammation and tissue damage. This increased vulnerability of neonates and infants can be attributed to the underdeveloped state of their gastrointestinal tract and the immaturity of their immune systems [4,6]. This susceptibility highlights the necessity of robust early immune defenses for a healthy gastrointestinal environment.

Breast milk is the premier source of nutrition for neonates, supplying essential nutrients and bioactive components indispensable for gastrointestinal and immune system maturation [7,8]. The presence of immunoglobulins, cytokines, oligosaccharides, and microbes in breast milk is recognized for bolstering an infant’s resistance to infections and nurturing the development of a healthy gut [9,10]. The composition of breast milk is influenced by a variety of maternal factors, including the mother’s diet and immune status, which can modulate the immunological components available to the infant [11,12].

The bioactive peptide surfactin, derived from *Bacillus* extracts, has emerged as a promising candidate for nutritional interventions that could modulate immune function [13,14]. Research has evidenced that surfactin can stimulate the innate immune response by activating dendritic cells, crucial components of the body’s frontline defense against pathogens [15]. However, the impact of maternal nutritional interventions with surfactin on the gut health of offspring remains to be fully elucidated.

Considering the well-documented advantages of breastfeeding, elucidating how maternal factors influence milk composition, and subsequently affect the offspring’s immune response is a critical research domain. This study aims to explore the impact of maternal consumption of surfactin on the immunological properties of breast milk and its effects on the neonatal gastrointestinal tract using mouse models and in vitro cell assays. This study also seeks to provide underlying strategies that could prevent early-life gut inflammatory injuries induced by enteric infections, thereby contributing to the advancement of nutritional strategies for infant health.

## 2. Materials and Methods

### 2.1. Animals and Bacterial Strains

Gravid mice (8–10 weeks old) were purchased from the Yangzhou University Laboratory Animal Center (Yangzhou, China). All the gravid mice and their offspring were from the C57BL/6 strain and kept and bred under the same specific-pathogen-free (SPF) conditions.

The *Salmonella typhimurium strain 1344* utilized in this study was maintained in a laboratory and cultured with agitation overnight at 37 °C in Luria–Bertani (LB) broth.

### 2.2. Extraction of Surfactin

*Bacillus subtilis OKB105* is a recombinant bacterium created through genome shuffling and maintained at the laboratory [15]. *Bacillus subtilis OKB105* was revived using LB media, and subsequently cultured in high quantity to form a seed culture. A total of 3% (*v*/*v*) seed culture was added to the Landy medium. The culture was then incubated at 33 °C at a rotating speed of 250 rpm for 36 h. The fermentation broth was centrifuged to extract the supernatant. The pH was then adjusted to 2.0, and a precipitate was produced by centrifugation after standing at 4 °C for 12 h. The precipitate was dissolved in ethanol, and the surfactin was produced using rotational evaporation. The concentration of surfactin was over 95%, as detected by HPLC.

### 2.3. Cell Culture

Mouse monocyte–macrophage RAW 264.7 cell lines were cryopreserved in our laboratory. DMEM medium supplemented with 10% fetal bovine serum and two antibiotics (100 U/mL penicillin and 100 µg/mL streptomycin sulfate) at 37 °C were used for the RAW 264.7 cells. The cells were incubated in an incubator (Thermo Fisher Scientific, Redmond, WA, USA) containing 5% CO_2_ until they reached approximately 80% confluence before treatment.

### 2.4. Experimental Design

#### 2.4.1. In Vivo

The Nanjing Agricultural University Institutional Animal Care and Use Committee authorized all methods and animal studies, which were carried out in accordance with the standards set out by the National Institute of Health.

A total of twenty-eight gravid mice were randomly categorized into two cohorts and were given surfactin or not in drinking water from one week after conception to 21 days postpartum. Namely, they were divided into the following two groups: 1. negative control group (M-NC, fed normally, n = 14); 2. surfactin group (M-SF, surfactin concentration in drinking water, 40 μg/mL, n = 14).

Cross-fostering experiments were conducted within 12 h after birth. The pups from the surfactin-supplemented dams were fostered and nursed by the control dams, while the pups from the control dams were nursed by the surfactin-supplemented dams. Namely, the mice were categorized into four groups: 1. negative control dams (n = 4) who fed the negative control pups (F-NC, n = 16); 2. negative control dams (n = 4) who fed the surfactin-treated pups (NC-SF, n = 16); 3. surfactin-treated dams (n = 4) who fed the negative control pups (SF-NC, n = 16); 4. surfactin-treated dams (n = 4) who fed the surfactin-treated pups (F-SF, n = 16). The pups were orally challenged with 1 × 10^9^ CFU *Salmonella typhimurium* at 21 days old, and then were slaughtered and sampled 48 h after infection.

#### 2.4.2. In Vitro

The mouse RAW 264.7 cells were pretreated with 100 µL of breast milk for 6 h, and then treated with LPS (1 µg/mL) for 24 h to induce RAW 264.7 cell inflammation (n = 3). Supernatants were collected for the further detection of nitric oxide (NO) and TNF-α. The total protein of the cells was determined by using RIPA Lysis Buffer (Biosharp, Beijing, China).

### 2.5. Sampling and Histology Detection

The gravid dams with nursing litters were separated from their pups for 6 h to allow for the accumulation of milk in the mammary glands. The dams were then anesthetized using 2% isoflurane (Sigma-Aldrich, St. Louis, MO, USA), and 4 U of oxytocin per mouse (MedChemExpress, Princeton, NJ, USA) was administered intraperitoneally between the left and right inguinal nipples to induce milk flow. Samples were collected using a Pasteur pipette that was modified to accommodate mouse nipples and to handle small liquid volumes. When no more milk was recovered from various nipples, milking was stopped, and the milk was stored at −20 °C until use [16].

After the experiment, the pups were euthanized for gross and histological examinations. Fragments of ileum tissue were fixed in 4% neutral buffered formalin for at least 24 h before being embedded in paraffin. Consecutive sections (5 μm thickness) were stained with hematoxylin and eosin. Histological lesions of the mice were quantified based on a prior investigation [17]. Briefly, the histology score ranged from 0 to 13 and was subdivided into the following categories: villus aspect (0 = normal, 1 = short, 2 = extremely short), villus tops (0 = normal, 1 = damaged, 2 = severely damaged), epithelium (0 = normal, 1 = flattened, 2 = damaged, 3 = severely damaged), inflammation (0 = no infiltration, 1 = mild infiltration, 2 = severe infiltration), crypts (0 = normal, 1 = mild crypt loss, 2 = severe crypt loss), crypt abscesses (0 = none, 1 = present), and bleeding (0 = none, 1 = present). For molecular studies, the remaining sections of the harvested intestinal samples were frozen in liquid nitrogen and kept at −70 °C.

### 2.6. Cytokines Detection

Ileum tissue was homogenized, and supernatants were collected. The cytokine concentrations in the ileum tissues (IL-1β, IL-6, TNF-α, and IL-10), the RAW 264.7 cells supernatants (TNF-α), and the breast milk (IL-4, IL-10, and TGF-β) were measured with ELISA kits (FineTest, Wuhan, China). The content of NO was determined using the Griess reagent kit assay (Beyotime, Shanghai, China).

### 2.7. RNA Isolation and Gene Expression

The total RNA from the ilea and the RAW 264.7 cells was extracted with Trizol (Vazyme, Nanjing, China) and quantified by spectrophotometry (NanoDrop ND1000, Thermo Scientific, Redmond, WA, USA). cDNA was synthesized by RNA according to instructions for the use of the manufacturer’s enzymes (Vazyme, Nanjing, China). A total of 1 μg of total RNA was reacted with a PrimeScript RT Reagent Kit (Vazyme, Nanjing, China) according to the manufacturer’s instructions.

Quantifications of the target genes claudin 1 (Cldn1), tight junction protein 1 (ZO1), occluding (Ocln), mucin 2 (Muc2), defensin 6 (Defa6), regenerating islet-derived 3 beta (Reg3b), interleukin 1 beta (IL-1β), interleukin 6 (IL-6), monocyte chemotactic protein 1 (MCP-1), Inducible Nitric Oxide Synthase (iNOS), interleukin 12 subunit p40 (IL-12p40), macrophage mannose receptor CD206 (CD206), transforming growth factor beta (TGF-β), arginase 1 (Arg-1), and a housekeeping gene (β-Actin) in the cDNA samples were carried out by fluorometric real-time PCR using a 7500 fluorescence detection system (Applied Biosystems, Carlsbad, CA, USA) and SYBR-Green PCR kits (Vazyme, Nanjing, China). The qPCR thermal cycling conditions were as follows: 30 s at 95 °C, followed by 40 cycles of 10 s at 95 °C and 30 s at 60 °C. The primers for individual genes are presented in Table 1. All the samples were tested in triplicate, and the gene expression levels were measured using the 2^−ΔΔCt^ method. The fold change value was computed for a gene expressed in the experimental vs. control conditions.

### 2.8. Western Blotting Assay

Total protein was collected by using RIPA Lysis Buffer (Biosharp, Beijing, China); SDS-PAGE separates the proteins of different sizes and electroporates them onto PVDF membranes (Millipore, Bedford, MA, USA). The membranes were blocked with 5% skimmed milk and incubated with anti-iNOS, anti-Arg1, and anti-β-actin antibodies at 4 °C overnight (Bioss, Beijing, China). After being washed with Tris-Buffered Saline with Tween 20 (TBST), the membranes were incubated with secondary antibodies for 2 h. Protein bands were visualized with an enhanced chemiluminescence (ECL) assay kit (Biosharp, Beijing, China) and measured with ImageJ software 1.53 k (NIH, Bethesda, MD, USA).

### 2.9. Immunofluorescence Assay

Tissue sections were deparaffinized twice in xylene, and then rehydrated in graded ethanol series, and washed with distilled water. Heat antigen retrieval was achieved using a microwave oven (Midea, Shunde, China) by incubating the slides in a citrate acid buffer solution (pH 6.0) at 96 °C for 20 min. After cooling, the ileum and mammary gland tissues were permeabilized with 0.5% Triton X-100 for 15 min and washed three times with PBS. Then, the tissues were incubated with 5% bovine serum albumin (BSA, Solarbio, Beijing, China) at 37 °C for 2 h to remove the nonspecific background. For IgA^+^ and CD3^+^ T cell staining, the cells were stained with primary antibodies (anti-IgA and anti-CD3 antibodies, 1:200, Abcam, Cambridge, UK) overnight at 4 °C. The samples were incubated with goat anti-rabbit and Alexa Fluor 488 (1:200, Abcam, Cambridge, UK) for 1 h at 37 °C, followed by DAPI for 8 min at room temperature. The samples were examined with a Zeiss 710 laser scanning confocal microscope (Carl Zeiss AG, Oberkochen, Germany). Fluorescence pictures were acquired, and the quantification of positive cells per unit area (0.6 mm^2^) was performed using ImageJ.

### 2.10. ROS Detection

For the determination of ROS production, the RAW 264.7 cells were treated with 10 μM 2′,7′-Dichlorodihydrofluorescein (DCFH-DA) (Beyotime, Shanghai, China) as per the manufacturer’s protocols, and fluorescence at 488/525 nm was detected using a Zeiss 710 laser scanning confocal microscope (Carl Zeiss AG, Oberkochen, Germany).

### 2.11. Antioxidant Enzyme Detection

Precooled normal saline was introduced into the ileum tissues of mice at a weight-to-volume ratio of 1:10. The homogenate was mechanically homogenized at 3000 rpm and centrifuged for 15 min under ice water bath conditions. The supernatant was taken, and according to the kit’s instructions (Solarbio, Beijing, China), the contents of SOD, MDA, and GSH-Px were determined.

### 2.12. Statistical Analysis

The results are reported as mean values plus or minus standard deviation (SD) and were analyzed using SPSS 17.0. One-way analysis of variance (ANOVA) was utilized to ascertain significant disparities among various the groups, while a t-test was used to determine the disparities between the two groups. The statistical significance levels are as follows: * *p* < 0.05 and ** *p* < 0.01. Unless otherwise specified, the data were aggregated from a minimum of three separate studies.

## 3. Results

### 3.1. Maternal Surfactin Administration Enhances Offspring Intestinal Development and Intestinal Innate Mucosal Immunity

The development of an offspring’s gut and immune system is largely influenced by maternal factors, including immune status and diet during pregnancy and lactation [18,19]. The gravid dams were administered surfactin via drinking water for 5 consecutive weeks, starting 2 weeks before delivery and continuing for 3 weeks postpartum (Figure 1A). The pups from the dams supplemented with surfactin showed no significant difference in weight on postnatal day 7 compared to those of the control group, which received only water. However, their weight significantly increased on days 14 and 21 (Figure 1B). Additionally, the pups from the surfactin-supplemented dams exhibited well-formed, longer villus and deeper crypt structures, along with the increased expression of Cldn1, with no significant difference in the ZO1 and Ocln contents (Figure 1C–H). Innate immunity, crucial for intestinal defense against pathogens and as a bridge to activate the adaptive immune system, was enhanced as evidenced by a significant upregulation of Muc2, Defa6, and Reg3b in the intestinal mucosa of the surfactin-supplemented pups (Figure 1I–K). Furthermore, the number of IgA^+^ and CD3^+^ T cells, key, innate immune cells, was significantly increased in the intestinal mucosa of these pups (Figure 1L,M).

### 3.2. Maternal Surfactin Administration Mitigates Intestinal Inflammatory Injury in Offspring

To determine whether the regulatory effects of surfactin on offspring occur during gestation (placental transfer) or postpartum (breast milk transfer), we conducted cross-fostering experiments. The pups from the surfactin-supplemented dams were fostered and nursed by the control dams (placental transfer), while the pups from the control dams were nursed by the surfactin-supplemented dams (breast milk transfer) (Figure 2A). The weight of the pups nursed by the surfactin-supplemented dams was significantly higher on postnatal day 21, regardless of whether they were biologically related or cross-fostered, compared to those nursed by the control dams. Notley, there was no significant difference in weight among the groups on initial postpartum day 7 (Figure 2B). To further evaluate whether the enhancing effects of surfactin on the offspring’s intestine could help resist intestinal infections and inflammation, the pups were infected with *Salmonella typhimurium*.

Compared with the pups nursed by the control dams, the pups nursed by the surfactin-supplemented dams exhibited significantly reduced intestinal tissue damage caused by *Salmonella typhimurium* infection, characterized by less necrosis and the detachment of the intestinal epithelium and reduced inflammatory cell infiltration (Figure 2C). Additionally, the expression levels of pro-inflammatory cytokines IL-1β, IL-6, and TNF-α were significantly reduced, while the expression of the anti-inflammatory cytokine IL-10 was promoted in the pups nursed by the surfactin-supplemented dams (Figure 2D–G). Regarding the oxidative stress levels, *Salmonella typhimurium* infection led to a significant decrease in SOD and GSH-Px levels and an increase in MDA levels in the intestines of the pups nursed by the control dams, whereas the surfactin-supplemented dams had significantly increased SOD and GSH-Px levels and a reduced MDA content in their nursed pups (Figure 2H–J). Collectively, these data suggest that surfactin-supplemented dams primarily exert regulatory effects on offsprings’ intestines through breast milk transfer rather than placental transfer, thereby aiding in resistance to *Salmonella typhimurium* infection-induced intestinal inflammatory injury.

### 3.3. Breast Milk from Surfactin-Fed Dams Ameliorates Offspring’s Inflammatory and Oxidative Stress via Macrophage Polarization Regulation

To explore the specific mechanisms by which the milk from the surfactin-supplemented dams (MSDs) modulates the immune response and suppresses infectious inflammation in the offspring’s intestinal mucosa, we isolated the milk and applied it to LPS-induced RAW macrophage inflammation models. The milk from both the control and surfactin-supplemented dams significantly reduced the levels of nitric oxide (NO) and TNF-α in the supernatant of LPS-induced macrophage cultures, with the MSDs showing a more pronounced inhibitory effect on these indicators (Figure 3A,B). Moreover, the MSD treatment more significantly inhibited the mRNA expression of IL-1β, IL-6, and MCP-1 induced by LPS in the RAW macrophages compared to that of the control group (Figure 3C–E). In terms of oxidative stress levels, the MSD treatment significantly suppressed the high levels of reactive oxygen species (ROS) induced by LPS, while the milk from the control dams had a less inhibitory effect on ROS (Figure 3F).

Macrophages are highly plastic and can polarize towards a multidimensional spectrum of phenotypes, including pro-inflammatory M1-like and anti-inflammatory M2-like states, in response to different local stimuli [20]. Microscopic observation revealed that LPS-induced inflammation in the RAW macrophages led to a morphological change from round or oval (M0) to polygonal (M1-like), consistent with the reported polarization of macrophages towards the pro-inflammatory M1 type. Interestingly, the MSD treatment reshaped the macrophage morphology, shifting from M1-like to spindle-shaped, indicative of an M2-like state (Figure 3G). Meanwhile, the milk from the control dams had no significant effect on macrophage morphology. Furthermore, after LPS induction, the expression levels of the M1-type-related IL-12p40 gene, the iNOS gene, and proteins were significantly increased (Figure 3H,I,M), while the expression levels of the M2-type-related CD206, the TGF-β genes, and the Arg-1 gene and protein were significantly decreased (Figure 3J–L,M). Importantly, the treatment for the MSDs reversed this phenomenon, suppressing the expression of M1-type genes and upregulating the expression of M2-type genes. In contrast, the milk from the control dams had a weaker regulatory effect on macrophage phenotype (Figure 3H–M).

### 3.4. Impact of Maternal Surfactin Feeding on the Content of Anti-Inflammatory Factors in Breast Milk

Anti-inflammatory cytokines were detected, and the results indicated that the levels of IL-4, IL-10, and TGF-β were significantly higher in the milk from the surfactin-supplemented dams (Figure 4).

## 4. Discussion

Maternal factors during pregnancy and lactation, including immune status, dietary habits, as well as breast milk composition, exert a profound influence on the susceptibility of offspring to early-life infectious inflammation and their overall lifelong health [16,21]. Interventions that bolster the innate immunity of offspring via maternal supplementation are critical for countering the risks associated with infectious and inflammatory diseases [3,7]. Bacillus subtilis produces a variety of secondary metabolites during fermentation, including surfactin, fengycin, and iturin. These metabolites exhibit anti-inflammatory and antibacterial properties, modulating both innate and adaptive immune responses. Among them, surfactin has the most significant effect [22,23,24,25]. Our previous studies have shown that surfactin can stimulate the innate immune response by activating dendritic cells [15]. While the extent of its impact on maternal immune status and milk composition through oral administration and its subsequent benefits to offspring are not fully understood, our findings shed light on its potential.

Healthy weight gain is an important indicator reflecting the nutritional status, intestinal function, and immune status of newborns [26,27]. Our results indicate that the offspring of the dams treated with surfactin exhibited faster weight gain, especially around the weaning period, but there was no significant difference in weight on the seventh day after birth. Since the transfer of maternal antibodies and other protective factors occurs through the placenta during pregnancy and via breast milk after birth, the above results suggest that the role of milk rather than placental transfer may account for the differences in weight gain. This aligns with the understanding that early life is a pivotal period for gut and immune system maturation, significantly influenced by maternal factors, especially breast milk [7,28]. Our analysis revealed enhanced intestinal development in the surfactin-supplemented offspring, characterized by a mature intestinal morphology and improved villus-crypt structures, indicative of enhanced nutritional absorption. Furthermore, the upregulation of genes related to the mucus barrier and antimicrobial peptides, as well as an increase in the number of IgA^+^ and CD3^+^ T cells, suggests that surfactin’s role in fortifying the neonatal mucosal immune barrier is crucial given the immature state of the neonatal immune system. These results are consistent with previous studies reporting the positive effects of bioactive components in breast milk, including IgA and human milk oligosaccharides (HMO), on the growth and gastrointestinal development of newborns [29,30].

Cross-fostering experiments are commonly used to determine whether maternal influences are transferred during pregnancy or lactation, that is, through placental or breast milk transfer [16,31]. In this study, cross-fostering showed that the offspring of the surfactin-supplemented dams had significantly higher body weights, regardless of biological relation or cross-fostering status, which indicating the significance of breast milk in mediating the benefits observed in surfactin-treated dams. A well-developed gut and its mucosal immune system are crucial defenses against intestinal pathogen infection [3]. *Salmonella typhimurium* was used to instigate intestinal infectious inflammation in the cross-fostered offspring; we assessed the capacity of surfactin to bolster intestinal development and mucosal immunity, thereby aiding in resistance to pathogen assault and curbing inflammatory sequelae. The results showed that the pups nursed by the surfactin-supplemented dams exhibited significantly reduced intestinal tissue damage caused by *Salmonella typhimurium* infection. This was characterized by enhancements in tissue morphology, the mitigation of inflammatory responses, and the attenuation of oxidative stress. These data suggest that the transfer of protective molecules via breast milk after birth, rather than placental transfer during pregnancy, is vital for protecting offspring. Similarly, a study confirmed through cross-fostering experiments that IgG transferred to nursing offspring via breast milk impeded intestinal infection [16]. Despite these insights, the precise constituents of the milk from the surfactin-treated dams that exert protective effects and how they function remain unclear.

Macrophages, integral to innate immune defense and immune homeostasis, exhibit phenotypes and functional states closely tied to intestinal environments [32,33]. Considering that the milk ingested by offspring may directly or indirectly affect the intestinal environment or interact with macrophages, in vitro experiments with milk and macrophages were conducted. We found that milk treatment significantly reduced the inflammatory response induced by LPS. Interestingly, we observed changes in macrophage morphology and significant differences among the different treatment groups. Based on the changes in cell morphology and the further detection of macrophage phenotype indicators, especially the gene and protein levels of the classic macrophage phenotype indicators Arg1 and iNOS, it was determined that the milk from the surfactin-treated dams induced a shift in macrophages from a pro-inflammatory M1-like state to an anti-inflammatory M2-like state. This finding is a novel aspect of our study. Such a shift in macrophage phenotype may be crucial for resolving inflammation and promoting the repair of neonatal intestinal tissue [20,34]. The literature indicates that the ability of breast milk to regulate macrophage polarization affects the development of neonatal immune responses and may contribute to long-term health outcomes [35]. In experimental animals such as pigs, the regulatory effect of milk on macrophages was also found in a study pointing out that the milk of sows fed fermented feed alleviated the decrease in Arg1 levels and increase in iNOS levels caused by LPS in colonic macrophages [36].

The cytokines in breast milk are important substances involved in the regulation of the offspring’s intestinal immunity [9,37]. Our study suggests that surfactin may affect the production of anti-inflammatory bioactive molecules in breast milk, with particularly significant increases in the IL-4, IL-10, and TGF-β contents in the milk of the surfactin-treated dams, which are known to be important molecules in regulating the shift of macrophages to an anti-inflammatory M2-like state [34]. Although we did not detect changes in all possible bioactive molecules with anti-inflammatory effects, our results partially explain the impact of surfactin on maternal mammary glands and milk, as well as the possible protective molecules transmitted through the mother and the mechanisms of action. Despite these promising results, future research should encompass a broader range of bioactive components, a long-term follow-up of offspring health, and the more comprehensive analysis of milk composition.

Our study represents a significant advancement in understanding how maternal nutritional interventions can influence neonatal health through breast milk. By demonstrating that the maternal intake of surfactin enhances neonatal intestinal development and immune modulation, we have identified a novel pathway through which the maternal diet can impact infant health. The use of cross-fostering experiments and in vitro assays provides robust evidence that surfactin exerts its beneficial effects primarily via breast milk, specifically through the modulation of macrophage polarization and the upregulation of anti-inflammatory cytokines. These findings underscore the potential of surfactin as a targeted nutritional strategy to improve early-life immune defenses and gastrointestinal health.

However, we acknowledge the limitations of our study. The reliance on mouse models means that the observed effects may not fully translate to humans, given the physiological differences between the species. Additionally, our focus on cytokines and macrophage polarization does not encompass the broader analysis of other bioactive components in breast milk that may also be influenced by surfactin. We agree that the comprehensive analysis of breast milk components, including metabolomics and microbiota, is necessary to fully elucidate the mechanisms underlying surfactin’s effects. Furthermore, clinical studies are essential to validate our findings in humans and to explore the long-term impact of surfactin supplementation on infant health.

In conclusion, while our study provides novel insights into the potential benefits of surfactin for neonatal health, we recognize the need for further research to address these limitations. We are currently planning additional studies to explore the broader effects of surfactin on breast milk composition and to conduct clinical trials that will help bridge the gap between our preclinical findings and human applications.

## 5. Conclusions

Our study provides evidence that the maternal consumption of surfactin can positively influence the immunological properties of breast milk and promote the health of the neonatal gastrointestinal tract. These findings contribute to the growing understanding of the role of maternal nutrition in infant health and suggest that targeted nutritional interventions during lactation may offer a viable strategy for enhancing early-life immune defenses.

## Figures and Tables

**Figure 1 nutrients-17-01009-f001:**
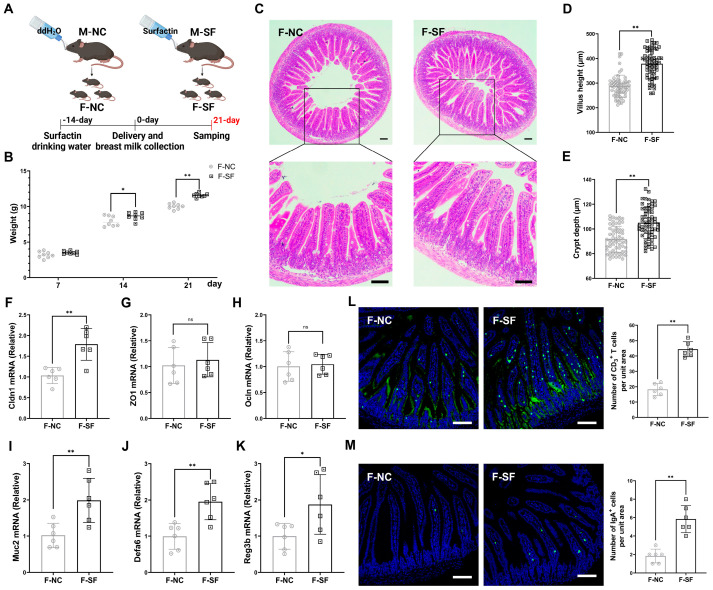
Maternal surfactin administration enhances offspring intestinal development and intestinal innate mucosal immunity. (n = 6) (**A**) Experimental design. Surfactin or not was given to gravid dams in drinking water for 5 weeks from 2 weeks pre-delivery to 3 weeks postpartum. (**B**) Body weight of pups was scored on days 7, 14, and 21. (**C**–**E**) Representative macroscopic images of intestinal morphology and bar graph individually show villus height and crypt depth. (**F**–**K**) mRNA levels of Cldn1, ZO1, Ocln, Muc2, Defa6, and Reg3b in ilea. (**L**,**M**) IgA and CD3 staining in ileum sections was observed using confocal microscopy, and bar graphs individually present density of IgA^+^ cells or CD3^+^ cells per unit area of mucosa. (F-NC: negative control dams fed negative control pups; F-SF: surfactin-treated dams who fed surfactin-treated pups). Scale bar is 100 μm. Data are expressed as mean ± SD (* *p* < 0.05; ** *p* < 0.01; *p* > 0.05 no significance).

**Figure 2 nutrients-17-01009-f002:**
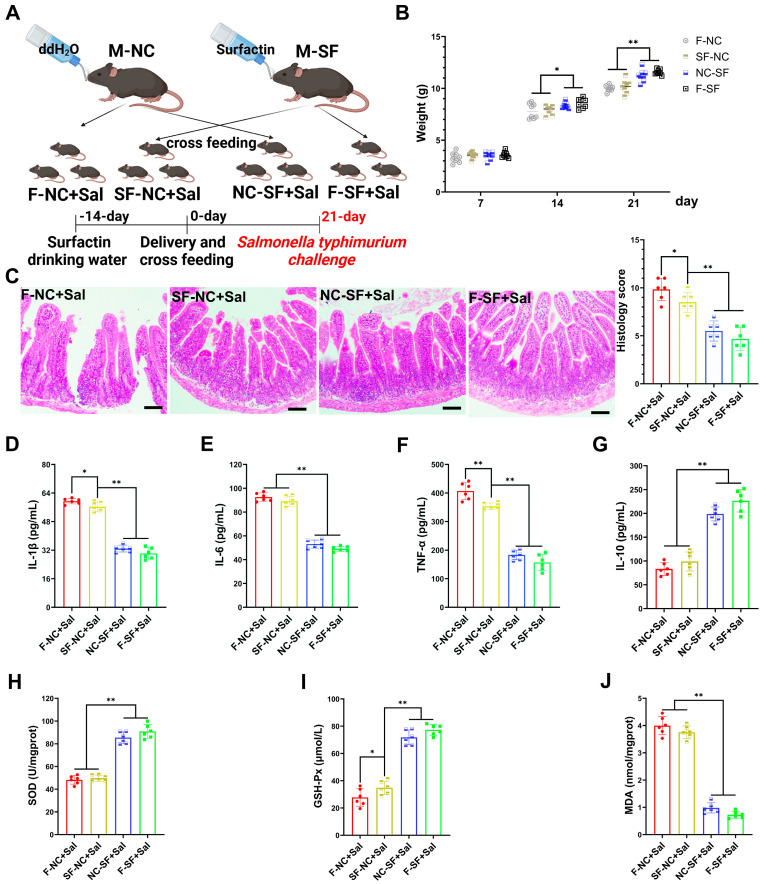
Maternal surfactin administration mitigates intestinal inflammatory injury in offspring. (n = 6) (**A**) Experimental design. 1. Negative control dams who fed negative control pups (F-NC); negative control dams who fed surfactin-treated pups (NC-SF); 3. surfactin-treated dams who fed negative control pups (SF-NC); 4. surfactin-treated dams who fed surfactin-treated pups (F-SF). Pups were challenged with 1 × 10^9^ CFU of *Salmonella typhimurium* (Sal) at 21 days. (**B**) Body weight of pups was scored on days 7, 14, and 21 before *Salmonella typhimurium* infection. (**C**) HE staining revealed histopathological changes in ileum tissues, and histologic scoring is detailed in Materials and Methods. Scale bar is 100 μm. (**D**–**G**) ELISA-detected levels of IL-1β, IL-6, TNF-α, and IL-10. (**F**–**J**) SOD, GSH-Px, and MDA concentrations in ileum tissues. Data are expressed as mean ± SD (* *p* < 0.05; ** *p* < 0.01).

**Figure 3 nutrients-17-01009-f003:**
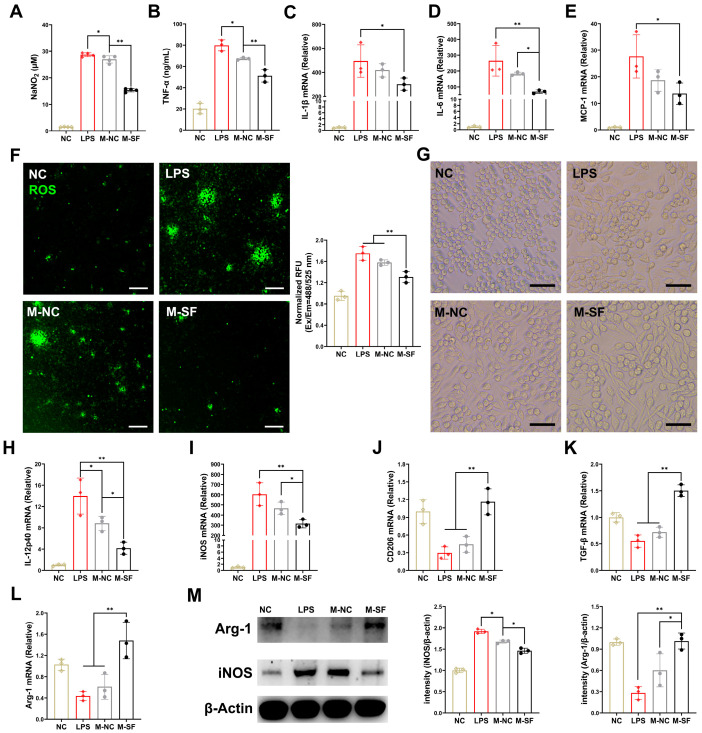
Breast milk from surfactin-fed dams ameliorates offspring’s inflammatory and oxidative stress via macrophage polarization regulation. (n = 3) (**A**,**B**) Concentrations of NO and TNF-α in macrophage supernatant. (**C**–**E**) mRNA levels of IL-1β, IL-6, and MCP-1 in macrophages. (**F**) Representative microscopy fluorescence images of ROS levels with DCF in macrophages and bar graph show quantification of ROS fluorescence intensity. (**G**) Representative macroscopic images of macrophages morphologies in response to different stimuli. (**H**–**L**) mRNA levels of IL-12p40, iNOS, CD206, TGF-β, and Arg-1 in macrophages. (**M**) Western blot for iNOS, arginase-1, and β-actin of control in macrophages with quantification of average across three separate experiments. Scale bar is 100 μm. Data are expressed as mean ± SD (* *p* < 0.05; ** *p* < 0.01).

**Figure 4 nutrients-17-01009-f004:**
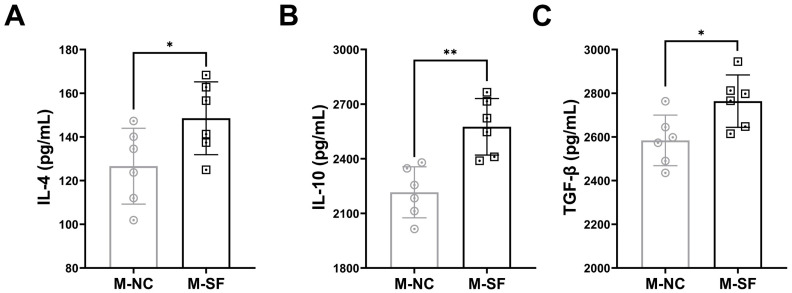
Impact of maternal surfactin feeding on content of anti-inflammatory factors in breast milk. (n = 6) (**A**–**C**) ELISA-detected levels of IL-4, IL-10, and TGF-β in breast milk. Data are expressed as mean ± SD (* *p* < 0.05; ** *p* < 0.01).

**Table 1 nutrients-17-01009-t001:** qPCR primer sequences.

Gene	Sequence (5′-3′)
Cldn1	F-GGGGACAACATCGTGACCG
	R-AGGAGTCGAAGACTTTGCACT
ZO1	F-ACCACCAACCCGAGAAGAC
	R-CAGGAGTCATGGACGCACA
Ocln	F-TTGAAAGTCCACCTCCTTACAGA
	R-CCGGATAAAAAGAGTACGCTGG
Muc2	F-TGACGTCTGGTGGAATGGTG
	R-CAGCGTAGTTGGCACTCTCA
Defa6	F-CCTTCCAGGTCCAGGCTGAT
	R-TGAGAAGTGGTCATCAGGCAC
Reg3b	F-ACTCCCTGAAGAATATACCCTCC
	R-CGCTATTGAGCACAGATACGAG
IL-1β	F-AGTTGACGGACCCCAAAAG
	R-TTTGAAGCTGGATGCTCTCAT
IL-6	F-CCAAGAGGTGAGTGCTTCCC
	R-CTGTTGTTCAGACTCTCTCCCT
MCP-1	F-AGCCAACTCTCACTGAAGCC
	R-GGACCCATTCCTTCTTGGGG
iNOS	F-GGAGTGACGGCAAACATGACT
	R-TCGATGCACAACTGGGTGAAC
IL-12p40	F-CGCCACACAAATGGATGCAA
	R-TGTGTCCTGAGGTAGCCGTA
CD206	F-CTCTGTTCAGCTATTGGACGC
	R-CGGAATTTCTGGGATTCAGCTTC
TGF-β	F-TTGGATTGCCAGTGCTAACCC
	R-AACAAGCCACAGTAACATGACA
Arg-1	F-CGTTGTATGATGCACAGCCG
	R-CCCCACCCAGTGATCTTGAC
β-Actin	F-GGCTGTATTCCCCTCCATCG
	R-CCAGTTGGTAACAATGCCATGT

## Data Availability

All data presented in this research are available through the corresponding author due to privacy, legal or ethical reasons.

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
