# Peer review of "Bacillus subtilis-Derived Surfactin Alleviates Offspring Intestinal Inflammatory Injuries Through Breast Milk"

_nutrients, 2025, doi:10.3390/nu17061009_

Round 1
Reviewer 1 Report
Comments and Suggestions for Authors
The manuscript “Bacillus subtilis-derived surfactin alleviates offspring intestinal inflammatory injuries through breast milk” provides information on the effect of surfactin on offspring intestinal inflammatory injuries through breast milk. Although the results are really good, the content needs to be improved to make it easier for readers to understand.
Major points
1. Are there any other factors identified from bacteria involved besides Sulfactin? It needs to add and discuss its information.
2. The authors examined various genes to verify the status of offspring intestinal inflammatory injury. A simple description is ok, it is necessary to describe what kind of gene it is.
3. Compared to humans, the authors need to add simple information about when intestinal damage occurs during the weaning period.
4. Based on the results of this study, do the authors think that people (Japanese, Chinese Korean, and so on) who often eat natto and kimchi would also observe the same effect of surfactin?
5. In this study, Surfactin was examined to evaluate the relationship between offspring intestinal inflammatory injury and breast milk. It is better to add a schematic model of some factors and how they are related to offspring intestinal inflammatory injury.
Comments on the Quality of English LanguageThe authors check the text of the manuscript, once again.
Author Response
Comments 1: Are there any other factors identified from bacteria involved besides Surfactin? It needs to add and discuss its information.
Response 1: We thank the reviewer for raising this question, which has helped us clarify our findings. This issue has been explored in the discussion section. “Bacillus subtilis produces a variety of secondary metabolites during fermentation, in-cluding surfactin, fengycin and iturin. These metabolites exhibit anti-inflammatory and antibacterial properties, modulating both innate and adaptive immune responses. Among them, surfactin has the most significant effect[23-26].”
Comments 2: The authors examined various genes to verify the status of offspring intestinal inflammatory injury. A simple description is ok, it is necessary to describe what kind of gene it is.
Response 2: Thank you for your valuable feedback. To better serve our readers, we have included detailed descriptions of the examined genes and their relevance to offspring intestinal inflammatory injury in the revised manuscript. We believe these details enhance the clarity and scientific rigor of our study.
Comments 3: Compared to humans, the authors need to add simple information about when intestinal damage occurs during the weaning period.
Response 3: Thank you for your insightful comment. We appreciate your suggestion to provide more context regarding the timing of intestinal damage during the weaning period in our study.
In our research, we focused on the weaning period in mice, which typically occurs around postnatal day 21. This period is critical for the development of the gastrointestinal tract and immune system in neonates. During this time, the transition from maternal milk to solid food can lead to significant changes in gut microbiota and intestinal physiology, making the gut more susceptible to inflammation and tissue damage. In humans, the weaning period is generally between 4 to 6 months of age, during which similar physiological changes and vulnerabilities can be observed.
We have added this information to the revised manuscript to provide a clearer context for our findings. Specifically, we have included a brief description in the Introduction section to highlight the importance of the weaning period in both mice and humans:
Revised Text (Introduction Section):
"Intestinal damage often occurs during the weaning period, a critical time for gastrointestinal development and immune maturation. In mice, this period typically occurs around postnatal day 21, while in humans, it corresponds to approximately 4 to 6 months of age. During this transition, the gut undergoes significant physiological changes, making it more susceptible to inflammation and tissue damage [5]."
We believe that this additional context will help readers better understand the relevance of our findings to both animal models and human health.
Thank you again for your valuable feedback. We have addressed all of your suggestions in the revised version of our manuscript.
Comments 4: Based on the results of this study, do the authors think that people (Japanese, Chinese Korean, and so on) who often eat natto and kimchi would also observe the same effect of surfactin?
Response 4: Thank you for raising this interesting and important question regarding the potential effects of surfactin in populations that frequently consume fermented foods like natto and kimchi.
Our Perspective: Based on the results of our study, we hypothesize that individuals who regularly consume natto, kimchi, and other fermented foods containing Bacillus subtilis (a producer of surfactin) may potentially experience similar beneficial effects on gut health and immune modulation. This hypothesis is supported by the following points:
1.Presence of Surfactin in Fermented Foods: Natto and kimchi, particularly those produced using traditional fermentation methods, often contain Bacillus subtilis, which is known to produce surfactin. Therefore, regular consumption of these foods could lead to increased intake of surfactin.
- Mechanistic Similarity: Our study demonstrates that surfactin can modulate the immune response and improve gut health by enhancing the anti-inflammatory properties of breast milk. Given the similar mechanisms of action, it is reasonable to hypothesize that surfactin from dietary sources could also exert beneficial effects on gut health in adults.
- Cultural Context: In populations such as Japanese, Chinese, and Korean individuals who frequently consume natto and kimchi, the regular intake of surfactin could potentially contribute to better gut health and reduced incidence of gastrointestinal inflammatory conditions.
Limitations and Considerations:
While our hypothesis is supported by the mechanisms observed in our study, several factors need to be considered:
- Dose-Dependence: The beneficial effects of surfactin may be dose-dependent. The amount of surfactin present in fermented foods can vary widely depending on the fermentation process and storage conditions. Therefore, the actual intake of surfactin from these foods may not always be sufficient to achieve the observed effects in our study.
- Human vs Animal Models: Our study was conducted using mouse models, and while the mechanisms are conserved across species, the direct translation of these findings to humans requires further validation. Clinical studies are needed to confirm the effects of surfactin in human populations.
- Dietary Interactions: The overall dietary context in which natto and kimchi are consumed may also influence the effects of surfactin. Other dietary components and lifestyle factors could interact with surfactin, potentially modulating its effects on gut health.
In summary, while our study suggests that surfactin has beneficial effects on gut health and immune modulation, further research is needed to confirm whether these effects can be observed in human populations that regularly consume natto, kimchi, and other fermented foods. We believe that our findings provide a strong rationale for future studies to explore the potential health benefits of surfactin in diverse cultural contexts.
Comments 5: In this study, Surfactin was examined to evaluate the relationship between offspring intestinal inflammatory injury and breast milk. It is better to add a schematic model of some factors and how they are related to offspring intestinal inflammatory injury.
Response 5: Thank you very much for your insightful suggestion regarding the inclusion of a schematic model to illustrate the relationship between surfactin, breast milk, and offspring intestinal inflammatory injury. We fully agree that such a visual representation would greatly enhance the clarity of our findings.
At present, due to time constraints, we have not yet prepared a detailed schematic model. However, we have included a brief textual description in the revised manuscript to outline the key factors and their relationships. This description aims to provide readers with a clearer understanding of how surfactin modulates breast milk composition and subsequently affects offspring intestinal health.
To provide a clearer understanding of the mechanisms explored in this study, we outline the key factors and their relationships as follows:
- Maternal Surfactin Consumption: Surfactin administered to dams enhances the levels of anti-inflammatory cytokines (e.g., IL-4, IL-10, TGF-β) in breast milk.
- Breast Milk Composition: These enhanced cytokines in breast milk contribute to the modulation of the neonatal immune response, promoting anti-inflammatory conditions in the offspring's gut.
- Offspring Intestinal Health: The presence of these anti-inflammatory factors in breast milk helps mitigate intestinal inflammation and promotes better intestinal development, as evidenced by improved villus-crypt structure and enhanced mucosal barrier function.
- Mechanistic Pathways: The observed effects are likely mediated through macrophage polarization from a pro-inflammatory (M1-like) to an anti-inflammatory (M2-like) phenotype, which is crucial for resolving inflammation and promoting tissue repair.
We apologize for the current omission of a schematic model and plan to include a detailed diagram in future revisions of the manuscript to visually represent these relationships.
Reviewer 2 Report
Comments and Suggestions for Authors
This is generally very useful research, which resulted in a good manuscript. The advantages of supplementing dams with surfactin (derived from Bacillus subtilis), via breast milk were demonstrated at many levels, in pups. The experiments were clear and Methods were described in details. Results are well organized and presented with attention to details, with the aid of illustrative and explanatory figures. The paragraph Discussion is beautifully conceived, with critical approach of their results. I have listed some minor comments, for consideration, below:
1.Title: “Bacillus subtilis-derived surfactin alleviates offspring intestinal inflammatory injuries through breast milk”. I suggest to make it clearer:
- It is about maternal supplementation with surfactin
- Also, that this is a study in mice (not humans)
2.Abstract:
- I would suggest making clear that this as research in mice (and not humans). I know the words “dams” and “pups” were used by the Authors, but just to make it clearer.
- Please insert in the Abstract: There were twenty-eight gravid mice (8-10 weeks old), divided in two groups and briefly present the intervention.
- Duration of the experiment – please insert in the Abstract
- Please replace “Mechanically…” – line 24.
3.Introduction
- Please update references - e.g. instead of 1, 2: “GBD 2016 Diarrhoeal Disease Collaborators. Estimates of the global, regional, and national morbidity, mortality, and aetiologies of diarrhoea in 195 countries: a systematic analysis for the Global Burden of Disease Study 2016. The Lancet Infectious Diseases. 2018.”
- Aim: “This study aims to explore the impact of maternal consumption of surfactin on the immunological properties of breast milk and its effects on the neonatal gastrointestinal tract and to provide underlying strategies that could prevent early-life gut inflammatory injuries induced by enteric infections, thereby contributing to the advancement of nutritional strategies for infant health.” From this aim, it is not clear at all that the research was performed in mice and also there was an in vitro study. Please revise.
4. Materials and Methods
“Namely, the mice were categorized into four groups: 1. Negative control dams fed negative control pups (F-NC, n = 16); 2. Negative control dams fed surfactin pups (NC-SF, n = 16); 3. Surfactin dams fed negative control pups (SF-NC, n = 16); 4. Surfactin dams fed surfactin pups (F-SF, n = 16). Please clarify how many pups/dams
5.Results
a.Maternal surfactin administration enhances offspring intestinal development and intestinal innate mucosal immunity
-I suggest removing sentences with references from Results and use them for Discussion.
-Figure 1: Please make all graphs related to F-NC more visible.
b.Maternal surfactin administration mitigates intestinal inflammatory injury in offspring
-Please make Figure 2B clearer.
c. Breast milk from surfactin-fed dams ameliorates offspring's inflammatory and oxidative stress via macrophage polarization regulation
- I suggest removing the sentence with references from Results and use it for Discussion.
- Figure 3. Please make graphs related to NC more visible. It is a pity, as figures are, otherwise, of high quality and illustrative.
d.Impact of maternal surfactin feeding on the content of anti-inflammatory factors in breast milk
- Same remark about the sentence with reference.
- Please insert place of “Figure 4” in the whole text.
6.Discussion
a.“Our previous studies have shown that surfactin can stimulate the innate immune response by activating dendritic cells”: please insert reference (14).
b.Please provide proper directions for future research.
c.Please emphasize the strength of your research and limitations.
7.Conclusion: Please revise “ical foundation for combating infections caused by other pathogenic microorganisms.”
8.References: Please insert more recent ones (e.g. - Jia J, Fu M, Ji W, Xiong N, Chen P, Lin J, Yang Q. Surfactin from Bacillus subtilis enhances immune response and contributes to the maintenance of intestinal microbial homeostasis. Microbiol Spectr. 2024 Oct 29;12(12):e0091824.)
9.ARRIVE Guidelines – Please revise lines reporting Item RESULTS (10)
10.I read attentively the iThenticate report and it looks fine.
Author Response
Comments 1. Title: “Bacillus subtilis-derived surfactin alleviates offspring intestinal inflammatory injuries through breast milk”. I suggest to make it clearer:
- It is about maternal supplementation with surfactin
- Also, that this is a study in mice (not humans)
Response 1: We thank the reviewer for raising this question. This study emphasizes the importance of breast milk in immune regulation, as highlighted in the title. Although the study was based on mouse models and in vitro cell tests, its implications are not limited to mice. Therefore, we plan to describe the treatment of dams and the results in the abstract.
Comments 2. Abstract:
- I would suggest making clear that this as research in mice (and not humans). I know the words “dams” and “pups” were used by the Authors, but just to make it clearer.
- Please insert in the Abstract: There were twenty-eight gravid mice (8-10 weeks old), divided in two groups and briefly present the intervention.
- Duration of the experiment – please insert in the Abstract
- Please replace “Mechanically…” – line 24.
Response 2: We are so grateful for your kind question. The abstract has been supplemented with experimental information on dams and pups. Replaced “Mechanically” with “In terms of mechanism”.
Comments 3. Introduction:
- Please update references - e.g. instead of 1, 2: “GBD 2016 Diarrhoeal Disease Collaborators. Estimates of the global, regional, and national morbidity, mortality, and aetiologies of diarrhoea in 195 countries: a systematic analysis for the Global Burden of Disease Study 2016. The Lancet Infectious Diseases. 2018.”
- Aim: “This study aims to explore the impact of maternal consumption of surfactin on the immunological properties of breast milk and its effects on the neonatal gastrointestinal tract and to provide underlying strategies that could prevent early-life gut inflammatory injuries induced by enteric infections, thereby contributing to the advancement of nutritional strategies for infant health.” From this aim, it is not clear at all that the research was performed in mice and also there was an in vitro study. Please revise.
Response 3: We thank the reviewer for raising this question. Some references in this study have been updated. The study aims have been revised to reflect the content of the research.
Comments 4. Materials and Methods:
“Namely, the mice were categorized into four groups: 1. Negative control dams (n = 4) fed negative control pups (F-NC, n = 16); 2. Negative control dams (n = 4) fed surfactin pups (NC-SF, n = 16); 3. Surfactin dams (n = 4) fed negative control pups (SF-NC, n = 16); 4. Surfactin dams (n = 4) fed surfactin pups (F-SF, n = 16). Please clarify how many pups/dams.
Response 4: We are so grateful for your kind question. The number of dams is described in Materials and Methods.
Comments 5. Results:
a.Maternal surfactin administration enhances offspring intestinal development and intestinal innate mucosal immunity
-I suggest removing sentences with references from Results and use them for Discussion.
-Figure 1: Please make all graphs related to F-NC more visible.
b.Maternal surfactin administration mitigates intestinal inflammatory injury in offspring
-Please make Figure 2B clearer.
- Breast milk from surfactin-fed dams ameliorates offspring's inflammatory and oxidative stress via macrophage polarization regulation
- I suggest removing the sentence with references from Results and use it for Discussion.
- Figure 3. Please make graphs related to NC more visible. It is a pity, as figures are, otherwise, of high quality and illustrative.
d.Impact of maternal surfactin feeding on the content of anti-inflammatory factors in breast milk
- Same remark about the sentence with reference.
- Please insert place of “Figure 4” in the whole text.
Response 5: We are so grateful for your kind question. a: To help readers better understand the background, content, and results of this key part of the research, a brief explanation is provided. b: High quality images have been uploaded. c: To facilitate the reader's understanding, a brief explanation is provided. High quality images have been uploaded. d: The deletion process has been completed. "Figure 4" has been inserted.
Comments 6. Discussion
- “Our previous studies have shown that surfactin can stimulate the innate immune response by activating dendritic cells”: please insert reference (14).
- Please provide proper directions for future research.
- Please emphasize the strength of your research and limitations.
Response 6: We are so grateful for your kind question. a: " reference (14)" has been inserted. b, c: Additional elements have been incorporated into the discussion.
Comments 7.Conclusion: Please revise “ical foundation for combating infections caused by other pathogenic microorganisms.”
Response 7: We are so grateful for your kind question. The deletion process has been completed.
Comments 8. References: Please insert more recent ones (e.g. - Jia J, Fu M, Ji W, Xiong N, Chen P, Lin J, Yang Q. Surfactin from Bacillus subtilis enhances immune response and contributes to the maintenance of intestinal microbial homeostasis. Microbiol Spectr. 2024 Oct 29;12(12):e0091824.)
Response 8: We are so grateful for your kind question. Some references in this study have been updated.
Comments 9.ARRIVE Guidelines – Please revise lines reporting Item RESULTS (10).
Response 9: We are so grateful for your kind question. The revisions have been completed as requested.
Comments 10.I read attentively the iThenticate report and it looks fine.
Response 10: Thank you very much for your detailed and helpful comments.
Reviewer 3 Report
Comments and Suggestions for Authors
Journal Nutrients (ISSN 2072-6643)
Manuscript ID nutrients-3463817
The study explores the effects of Surfactin, a bioactive peptide extracted from Bacillus extracts, on neonatal health via its supplementation to the gravid mice. The study highlights that Surfactin, through maternal breast milk, improved the pups' body weight, intestinal health, and immunity. Cross-fostering experiments also indicated that the pups exposed to Salmonella infection showed enhanced immunity, lesser intestinal damage, and inflammation. The expression of proinflammatory markers was suppressed with milk and promoted a shift in macrophage polarization from the M1 to the M2 state. The study further highlights that Surfactin supplementation to gravid mice influences the composition of breast milk and results in the improved health of the offspring. The study highlights the importance of nutritional strategies that can be employed to improve early-life immunity and gut health.
The study is fascinating and well-explained. The study has a very distinct research background. However, I have a few suggestions for the authors:
- Please check the word offerings in line 64, please modify lines 64-65, “All gravid mice and their offerings were in the C57BL/6 background and bred/kept under the same specific-pathogen-free (SPF) conditions.” This can be written as: All the gravid mice and their offspring were from the C57BL/6 strain and kept and bred under the same specific-pathogen-free (SPF) conditions.
- I observed spacing between a number and the units in some places, for example, lines 743 and 76 but not in some places e.g. lines 76 and 78. It is suggested to keep the space between the unit and numbers and keep it consistent throughout the draft.
- Please correct the spellings of height in graph D in Figure 1.
- Please indicate what F-NC and F-SF stand for, at the end of the legend of Figure 1.
- Always indicate (n = 6) or n=3, at the end of the figure legends.
- Please write as data are expressed/represented as mean ±SD, please change it in the figure legends.
- Lines 398-399 are not clear; please rewrite them for better understanding. Line 259: The challenged word makes no clear meaning; please replace it with a suitable word.
I accept the paper after some minor revisions/suggestions.
Author Response
The study explores the effects of Surfactin, a bioactive peptide extracted from Bacillus extracts, on neonatal health via its supplementation to the gravid mice. The study highlights that Surfactin, through maternal breast milk, improved the pups' body weight, intestinal health, and immunity. Cross-fostering experiments also indicated that the pups exposed to Salmonella infection showed enhanced immunity, lesser intestinal damage, and inflammation. The expression of proinflammatory markers was suppressed with milk and promoted a shift in macrophage polarization from the M1 to the M2 state. The study further highlights that Surfactin supplementation to gravid mice influences the composition of breast milk and results in the improved health of the offspring. The study highlights the importance of nutritional strategies that can be employed to improve early-life immunity and gut health.
The study is fascinating and well-explained. The study has a very distinct research background. However, I have a few suggestions for the authors:
Comments 1: Please check the word offerings in line 64, please modify lines 64-65, “All gravid mice and their offerings were in the C57BL/6 background and bred/kept under the same specific-pathogen-free (SPF) conditions.” This can be written as: All the gravid mice and their offspring were from the C57BL/6 strain and kept and bred under the same specific-pathogen-free (SPF) conditions.
Response 1: We thank the reviewer for raising this question. The requested modifications have been made accordingly.
Comments 2: I observed spacing between a number and the units in some places, for example, lines 743 and 76 but not in some places e.g. lines 76 and 78. It is suggested to keep the space between the unit and numbers and keep it consistent throughout the draft.
Response 2: We are so grateful for your kind question. The full text has been thoroughly checked.
Comments 3: Please correct the spellings of height in graph D in Figure 1.
Response 3: We thank the reviewer for raising this question. Figure 1 has been modified accordingly.
Comments 4: Please indicate what F-NC and F-SF stand for, at the end of the legend of Figure 1. Always indicate (n = 6) or n=3, at the end of the figure legends. Please write as data are expressed/represented as mean ±SD, please change it in the figure legends.
Response 4: We are so grateful for your kind question. The figure legends have been revised as requested.
Comments 5: Lines 398-399 are not clear; please rewrite them for better understanding. Line 259: The challenged word makes no clear meaning; please replace it with a suitable word.
Response 5: We thank the reviewer for raising this question. Lines 398-399 have been deleted as requested; The experimental design has been further detailed to address the feedback.
I accept the paper after some minor revisions/suggestions.
We are very grateful for your insightful suggestions and have addressed them in the revised manuscript
Round 2
Reviewer 1 Report
Comments and Suggestions for Authors
I think that the revised manuscript has been improved.